# A multi-agent approach to neurological clinical reasoning

Moran Sorka[1,2], Alon Gorenshtein[1,4], Dvir Aran[2,3☯*], Shahar Shelly[1,4,5☯*]

**1** AI Neurology Laboratory, Ruth and Bruce Rapaport Faculty of Medicine, Technion-Institute of Technology, Haifa, Israel, **2** Faculty of Biology, Technion-Israel Institute of Technology, Haifa, Israel, **3** The Taub Faculty of Computer Science, Technion-Israel Institute of Technology, Haifa, Israel, **4** Department of Neurology, Rambam Health Care Campus, Haifa, Israel, **5** Department of Neurology, Mayo Clinic, Rochester, Minnesota, United States of America

☯ These authors contributed equally to this work.
* dviraran@technion.ac.il (DA); s_shelly@rmc.gov.il (SS)

## Abstract

Large language models (LLMs) have demonstrated impressive capabilities in medical domains, yet their ability to handle the specialized reasoning patterns required in clinical neurology warrants systematic evaluation. Neurological assessment presents distinctive challenges that combine anatomical localization, temporal pattern recognition, and nuanced symptom interpretation—cognitive processes that are specifically tested in board certification examinations. We developed a comprehensive benchmark comprising 305 questions from Israeli Board Certification Exams in Neurology and classified each along three dimensions of complexity: factual knowledge depth, clinical concept integration, and reasoning complexity. We evaluated ten LLMs of varying architectures and specializations using this benchmark, testing base models, retrieval-augmented generation (RAG) enhancement, and a novel multi-agent system. Our analysis revealed significant performance variation across models and methodologies. The OpenAI-o1 model achieved the highest base performance (90.9% accuracy), while specialized medical models performed surprisingly poorly (52.9% for Meditron-70B). RAG enhancement provided variable benefits across models; substantial improvements for mid-tier models like GPT-4o (80.5% to 87.3%) and smaller models, but limited effectiveness on the highest complexity questions regardless of model size. In contrast, our multi-agent framework—which decomposes neurological reasoning into specialized cognitive functions including question analysis, knowledge retrieval, answer synthesis, and validation—achieved dramatic improvements, especially for mid-range models. The LLaMA 3.3-70B-based agentic system reached 89.2% accuracy compared to 69.5% for its base model, with particularly substantial gains on level 3 complexity questions across all dimensions. External validation on MedQA revealed dataset-specific RAG effects: while RAG improved board certification performance, it showed minimal benefit on MedQA

**Data availability statement:** All data are available as supplemental tables.

**Funding:** The author(s) received no specific funding for this work.

**Competing interests:** The authors have declared that no competing interests exist.

questions (LLaMA 3.3-70B: + 1.4% vs + 3.9% on board exams), reflecting alignment between our specialized neurology textbook and board examination content rather than the broader medical knowledge required for MedQA. Most notably, the multi-agent approach transformed inconsistent subspecialty performance into remarkably uniform excellence, effectively addressing the neurological reasoning challenges that persisted even with RAG enhancement. We further validated our approach using an independent dataset comprising 155 neurological cases extracted from MedQA. The results confirm that structured multi-agent approaches designed to emulate specialized cognitive processes significantly enhance complex medical reasoning offering promising directions for AI assistance in challenging clinical contexts.

## Author summary

Our research addresses a critical question in artificial intelligence for healthcare: can language models effectively handle the complex reasoning required in neurological diagnosis? We developed a comprehensive benchmark based on actual neurology board certification questions and evaluated how various AI models perform on these challenging cases. While base language models showed promising capabilities, they struggled with the most complex neurological scenarios. Significantly, we discovered that simply providing these models with access to medical textbooks through retrieval-augmented generation (RAG) offers only modest improvements. The breakthrough in our work was the development of a novel multi-agent framework that decomposes complex neurological reasoning into specialized cognitive functions distributed across five distinct agents – mirroring how expert neurologists approach difficult cases. This structured approach dramatically outperformed both base models and RAG-enhanced systems, transforming inconsistent performance into remarkably uniform excellence across neurological subspecialties. Most impressively, our multi-agent system enabled mid-tier models to achieve near-expert level performance, with the open-sourced LLaMA 3.3-70B model improving from 69.5% to 89.2% accuracy. Our framework demonstrates that structured reasoning architectures can significantly enhance AI systems in complex clinical contexts, offering a promising direction for medical AI that complements rather than replaces human expertise.

## Introduction

Recent advances in large language models (LLMs) have demonstrated remarkable capabilities in medical reasoning tasks [1–3], with models such as GPT-4 exceeding medical professionals on standardized examinations [4] and handling diagnostic challenges across various medical specialties [5–9]. Recent studies assessed LLMs potential on neurology; Schubert et al. [7] used board-style examinations and

Nógrádi et al. [8] evaluated ChatGPT on neurology-specific specialist exams showing promising diagnostic augmentation performance. However, significant questions remain about how these models handle distinctive cognitive challenges of neurological reasoning, particularly complex integration tasks requiring detailed anatomical knowledge, temporal pattern recognition, and synthesis across multiple neural systems.

To address these questions, we focus on neurology as it presents distinctive challenges that provide an excellent opportunity to evaluate the advanced reasoning capabilities of current LLMs. The field requires integration of detailed anatomical knowledge, temporal pattern recognition, and synthesis of diverse clinical presentations that may span multiple neural systems. Studies have shown that neurologists consistently see patients with higher markers of complexity compared to many other medical specialties, highlighting the inherently challenging nature of the field [10]. Neurological board certification examinations test these multifaceted reasoning skills, requiring candidates to analyze complex clinical scenarios where diagnostic and management decisions depend on recognizing subtle patterns, determining precise anatomical localization, and considering the temporal evolution of symptoms. Such questions demand more than factual recall - they require the ability to systematically evaluate competing hypotheses while managing uncertainty and integrating findings across different domains. This form of assessment, which mirrors the cognitive processes employed in actual clinical practice, provides an ideal benchmark for evaluating whether current LLMs can effectively navigate the type of structured reasoning tasks that characterize expert medical decision-making.

While retrieval-augmented generation (RAG) enhances LLM performance in specialized domains [11–13], neurological assessment may demand more sophisticated reasoning architectures. Recent advances in LLM-based agentic systems, including multi-agent collaboration frameworks and tool-augmented reasoning models, offer a promising approach for decomposing complex tasks into specialized cognitive functions that mimic clinical experts' structured problem-solving approach [14,15]. These systems can reason, plan, and collaborate to complete intricate reasoning tasks [16–18]. Goodell et al. [19] showed that equipping LLMs with specialized medical calculation tools (an agentic approach) reduced calculation errors by >10× compared to un-augmented models. With multi-agent systems like Med-Chain, demonstrating significant improvements over single-agent models [20].

Multi-step reasoning is a focus beyond medicine. In finance, the FinBen benchmark [21] includes multi-step numerical and decision-making tasks, finding that even top LLMs excel at data extraction but "struggle with advanced reasoning and complex tasks" like forecasting, and Egg et al. [22] created DABstep, comprising hundreds of multi-stage financial data challenges requiring iterative code-based analysis. Those works illustrate the growing interest and challenges of applying AI agents to multi-step financial reasoning.

We address these challenges through three main contributions: (1) developing a comprehensive benchmark for evaluating LLM performance in neurological assessment, based on board certification questions; (2) conducting systematic evaluation of current LLMs on this benchmark, including testing RAG enhancement efficacy; (3) introducing a novel multi-agent framework that decomposes complex neurological reasoning into specialized cognitive functions, demonstrating significant performance improvements beyond base models or standard RAG systems. This structured, agentic approach offers a promising direction for addressing complex medical reasoning challenges while maintain analytical rigor essential for neurological assessment.

## Methods

### Multiple-choice question dataset

We analyzed 305 multiple-choice questions (MCQs) from Israeli Board Certification Exams in Neurology (June 2023-September 2024). These examinations assess physician's depth of knowledge, clinical reasoning, and decision-making abilities through clinical vignettes requiring integration of information from neuroanatomy, pathophysiology, imaging, and treatment guidelines. Questions emphasize case-based reasoning with ambiguous scenarios, forcing candidates to prioritize differential diagnoses, weigh risk factors, and manage decisions. Each MCQ presented a clinical scenario with

four answer options, with a passing score threshold of 65% required for board certification. Questions were professionally translated from Hebrew to English, excluding those containing visual elements to ensure compatibility with current language model limitations.

### Classifying questions into neurological subspecialties and validation

All MCQs were categorized into 13 neurological subspecialties by two authors (SS and AG) capturing the main neurological field in the question, if two fields were involved, we classified the question according to the dominant diagnostic domain. A panel of senior neurologists validated question-answer pairs against current clinical guidelines and classified them based on reasoning complexity.

### Validation dataset

To validate our findings beyond the Israeli Board Certification benchmark, we utilized the MedQA dataset [23] as an independent validation corpus. From the original 1,273 questions in the dataset, we identified and extracted 155 questions (12.1%) specifically related to neurological topics and conditions. The selection of neurological questions was performed through an initial classification using an LLM, followed by thorough manual validation by two authors (SS and AG) to ensure clinical relevance and accuracy of categorization. The MedQA neurological subset provided an important complementary evaluation resource as it originated from a different source (US medical licensing examinations) and exhibited different question structures and emphasis compared to our primary benchmark.

### Statistical analysis

We used accuracy as our primary metric, accompanied by 95% confidence intervals (Wilson score method), with F1 scores providing a balanced precision-recall measurement. Statistical significance between implementation approaches was determined using Fisher's exact test (nominal $\alpha = 0.05$ threshold). Given the exploratory nature of multiple pairwise comparisons (13 total: 10 base-vs-RAG, 3 base-vs-agents), we report unadjusted p-values, recognizing that findings should be interpreted as hypothesis-generating rather than confirmatory. Pearson's correlation analyzed relationships between our three complexity categories. Implementation used Python 3.11.9, with crewai (version 0.95.0) for multi-agent system development, chromadb (version 0.5.10) for vector database management, and ollama (version 0.4.4) for local model.

### Multi-agent framework

We developed an agentic system using the CrewAI framework to simulate clinical neurological reasoning through five specialized agents (Fig 1). The *Question Complexity Classifier* initiates the process by analyzing the clinical scenario and categorizing it based on reasoning complexity, distinguishing between questions requiring simple fact recall versus those demanding integration of multiple clinical concepts and diagnostic reasoning. This initial classification guides subsequent processing strategies.

The *Question Interpreter* systematically decomposes questions into key medical concepts, identifying critical diagnostic elements, relevant symptoms, patient history, and other clinical factors. This agent generates optimized search queries for each identified concept to ensure comprehensive knowledge retrieval, effectively transforming clinical scenarios into structured data representations.

The *Research Retrieval Agent* interfaces with our RAG system to gather relevant neurological knowledge. Due to inherent token limitations of LLMs (Table A in S1 File), we implemented a file-based information persistence strategy. For each generated query, the agent searches through the knowledge base and systematically stores retrained passages using a SaveFile tool. This approach preserves the full context and relationships between multiple retrieved passages that

PLOS Digital Health

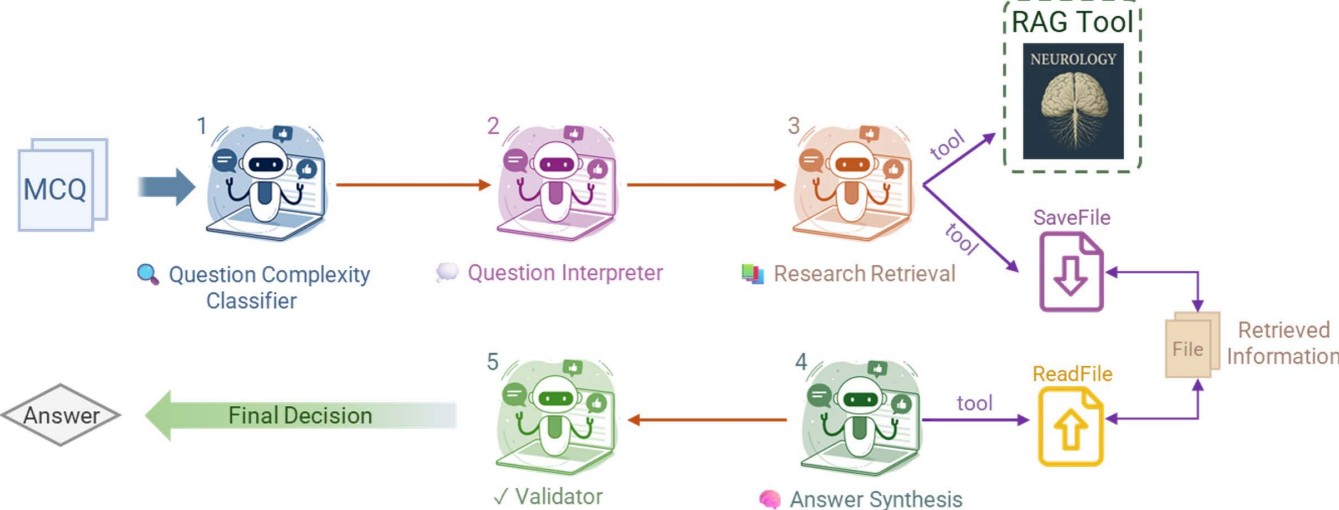

**Fig 1. A Multi-agent system workflow for neurology question answering.** This figure illustrates a comprehensive multi-agent system architecture designed for processing and answering multi-choice Neurology questions (MCQs). The system comprises five specialized agents: Question Complexity Classifier (1), Question Interpreter (2), Research Retrieval agent (3), Answer Synthesis agent (4) and Validator agent (5). Icons were generated with OpenAI's ChatGPT image generation tool. Under the OpenAI Terms of Use, users own the generated content.

would otherwise be lost due to token constraints. The agent performs multiple retrieval rounds based on the decomposed queries, accumulating comprehensive context for the clinical scenario.

The *Answer Synthesis Agent* processes multiple sources through sequential reading operations: the original MCQ, retrieved knowledge chunks, and the decomposed clinical concepts. While the file-based storage preserves all retrieved information, the agent works within model context limitations by accessing stored knowledge incrementally via a ReadFile tool, processing information in manageable chunks while maintaining reasoning coherence. The agent employs structured reasoning to evaluate each potential answer option, synthesizing evidence from processed knowledge chunks to construct justified responses.

The *Validator Agent* serves as the final quality assurance checkpoint, evaluating synthesized answers against established medical criteria and knowledge base consistency. If responses meet validation thresholds, they proceed as final output. When discrepancies are identified, the Validator initiates a feedback loop to the *Question Interpreter*, triggering a refined analysis cycle to ensure diagnosis. The entire framework maintains JSON-structured data flow through a centralized controller, ensuring proper sequential execution and data integrity – enabling comprehensive knowledge integration while managing model token limitations through efficient file handling.

## Experiments evaluation

We evaluated multiple commercial and open-source LLMs using standardized prompts (temperature 0.7, top-p 0.9). We selected temperature 0.7 to balance factual accuracy with response naturalness, following established practices in medical question answering [24,25], where temperatures ≤0.7 minimize hallucinations while maintaining response quality. Top-p 0.9 further constrains sampling to high-probability tokens, enhancing output coherence. No model-specific hyperparameter tuning was performed to ensure fair cross-model comparison. Models spanned three categories: general-purpose (OpenAI GPT-4 family, LLaMA), reasoning-specialized (OpenAI-o1, DeepSeek R1), and medical-domain (OpenBioLLM, MedLLaMA3, Meditron), with parameter scales from 8B to 70B. Model versions were: OpenAI o1-preview (2024-12-01-preview), GPT-4o and GPT-4o-mini (2024-08-01-preview), LLaMA 3.3-70B-Instruct (December 6, 2024),

LLaMA 3.1-8B-Instruct (July 23, 2024), DeepSeek-R1-Distill-Llama-70B and DeepSeek-R1-Distill-Llama-8B (January 2025), Meditron-70B, OpenBioLLM-70B, and MedLLaMA3 v20. Models spanned three categories: general-purpose (OpenAI GPT-4 family, LLaMA), reasoning-specialized (OpenAI-o1, DeepSeek R1), and medical-domain (OpenBioLLM, MedLLaMA3, Meditron), with parameter scales from 8B to 70B.

Our RAG framework used Bradley and Daroff's Neurology in Clinical Practice (8th edition, 2022) [26] as the knowledge base, processed through semantic chunking (512 tokens with 128 overlap) and embedded using BAAI's bge-large-en-v1.5 model (normalized embeddings, batch size 32). The vector database (ChromaDB version 0.5.10) retrieved the top 60 most similar passages per query using cosine similarity. After RAG evaluation, we selected top-performing models (o1, GPT-4o, LLaMA-70B) for testing within our agentic framework.

## Results

### Characteristics of the neurological assessment benchmark

Our benchmark comprises 305 board-level questions derived from Israeli Board Certification Exams in Neurology [27] across three recent examination periods (Fig 2a and S2 File). Questions span 13 neurological subspecialties, with Neuromuscular disorders (22%), Behavioral & Cognitive Neurology (15%) and Movement Disorders (10%) most represented

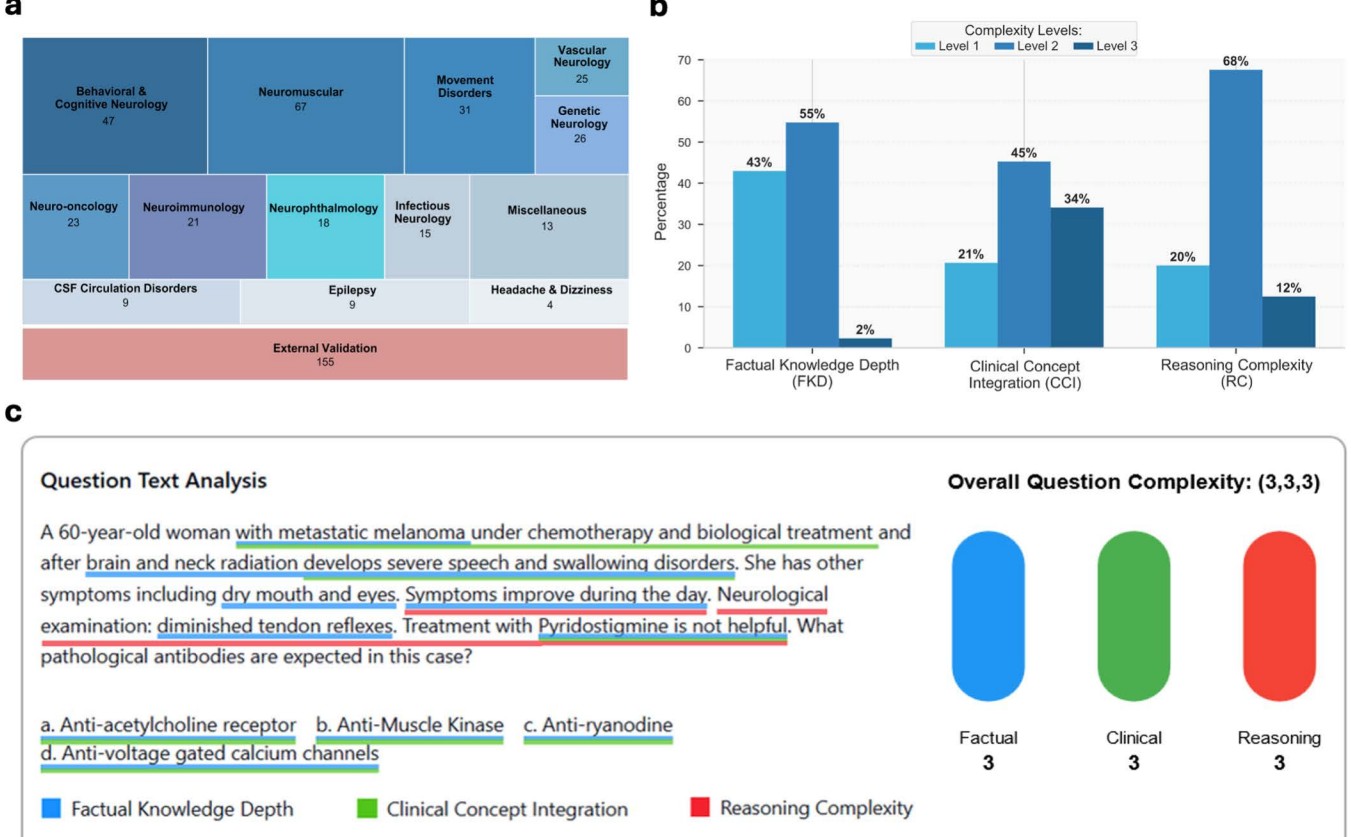

**Fig 2. Neurological Assessment Benchmark Composition and Complexity Framework.** This figure illustrates (a) Distribution of the 305 neurological board certification questions across 13 subspecialties. (b) Distribution of complexity levels across the three dimensions: factual knowledge depth (FKD), clinical concept integration (CCI), and reasoning complexity (RC). (c) An example question from the benchmark illustrating the multidimensional reasoning required in neurological assessment.

(Table B in S1 File). We classified each question by reasoning type: 263 questions (86%) required diagnostic reasoning such as syndrome identification, localization, and differential diagnosis, while 42 questions (14%) required therapeutic reasoning including treatment selection and management approaches (S2 File).

We developed a three-dimensional classification framework (Fig 2b), with each question categorized according to three distinct dimensions of complexity: factual knowledge depth (FKD), assessing the specialization level of medical knowledge required, from basic medical education to subspecialty expertise; clinical concept integration (CCI), measuring the number of clinical concepts that must be simultaneously considered, from single concept application to integration of three or more concepts with numerous variables; and reasoning complexity (RC), evaluating the sophistication of reasoning required, ranging from straightforward "if-then" logic to advanced temporal, probabilistic, or multi-step reasoning with management of contradictions and uncertainties. Correlation analysis between dimensions revealed moderate relationships (FKD-CCI: 0.56, FKD-RC: 0.51, CCI-RC: 0.67), confirming they capture distinct aspects of question complexity.

A representative example of question complexity is illustrated in Fig 2c, which demonstrates a case involving a 60-year-old woman with metastatic melanoma who develops severe speech and swallowing disorders after brain and neck radiation. This question exemplifies high complexity across our measurement framework, scoring level 3 (L3) for FKD - requiring specialized knowledge of neurological complications of cancer treatment and antibody-mediated disorders, L3 for CCI - requiring integration of multiple clinical concepts including radiation effects, autoimmune mechanisms, and neuromuscular junction pathophysiology, and L3 for RC - demanding sophisticated reasoning to differentiate between treatment side effects and paraneoplastic syndromes. With a composite complexity score of 9, this question falls in the highest category of overall complexity within our benchmark. The question demonstrates the multi-dimensional reasoning required in neurological diagnosis, particularly the need to recognize patterns suggesting myasthenia gravis or Lambert-Eaton syndrome in cancer patients with characteristic symptoms that fluctuate during the day and respond poorly to specific treatments.

## Base model performance

We evaluated ten large language models of varying architectures and parameter scales (Table A in S1 File). The performance revealed a clear hierarchy (Fig 3a and Table 1): the reasoning-specialized OpenAI-o1 model emerged as the strongest performer, achieving 90.9% accuracy (95% CI: 87.1-93.6%) across the full benchmark, significantly exceeding the threshold typically required for board certification. The general-purpose GPT-4o followed with 80.5% accuracy (95% CI: 75.9-84.7%), while DeepSeek-R1-70B achieved 87.7% accuracy (95% CI: 83.4-90.8%). LLaMA 3.3-70B performed in the mid-range with 69.5% accuracy (95% CI: 64.1-74.4%). Notably, the medical domain-specialized models performed below expectations, with OpenBioLLM-70B reaching 65.9% accuracy (95% CI: 60.4-71.0%) and Meditron-70B achieving only 52.9% accuracy (95% CI: 47.2-58.3%), suggesting that domain specialization alone does not confer advantages in complex reasoning. Performance remained relatively consistent across the three examination periods, with minor variations that did not alter the overall ranking pattern (Table B in S1 File).

When analyzed through our three-dimensional complexity framework, base model performance showed clear degradation patterns with increasing complexity levels. For the top-performing o1 model, performance remained generally strong but showed relative weaknesses in CCI L3 (86.5%) and FCD L3 (87.5%) while maintaining higher performance on RC tasks. The mid-range GPT-4o showed more pronounced degradation, particularly for FCD L3 (70.0%) and CCI L3 (75.7%). The pattern was most evident in the LLaMA 3.3-70B model, where accuracy for L3 complexity dropped substantially: 57.5% for FKD L3, 66.3% for CCI L3, and 61.0% for RC L3 (Fig 3b).

To validate our findings beyond the Israeli Board Certification benchmark, we evaluated our top-performing models on neurological questions extracted from the MedQA dataset. Among 1,273 total MedQA questions, we identified 155 neurological questions, providing a complementary evaluation dataset. Consistent with our board certification results, base model performance followed similar patterns, with o1 achieving the highest accuracy (96.8%) (96.8%, 95% CI: 92.7-98.6%), followed by GPT-4o (85.2%, 95% CI: 78.7-89.9%) and LLaMA 3.3-70B (76.8%, 95% CI: 69.5-82.7%) (Table D in S1 File).

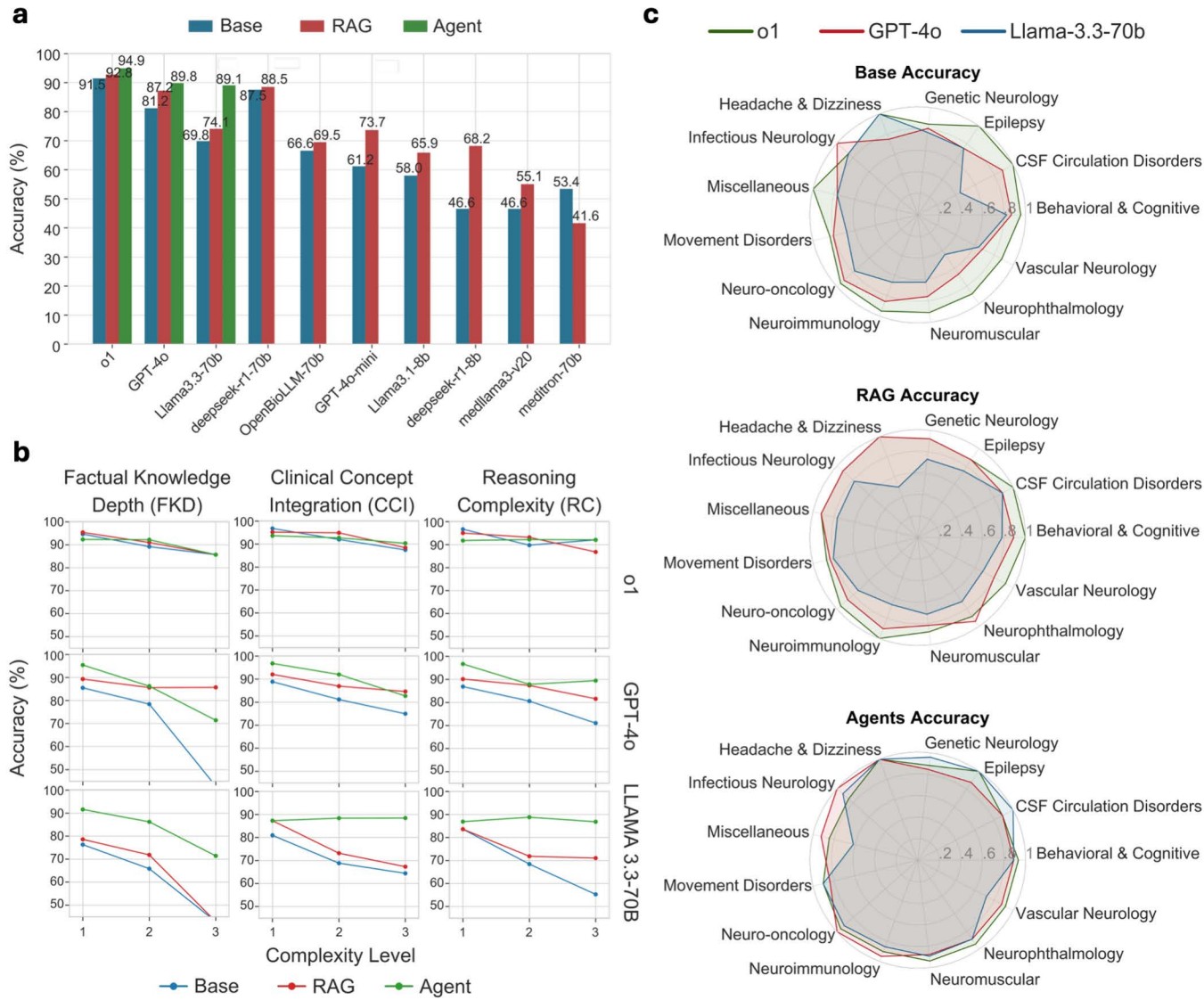

**Fig 3. Performance analysis of large language models in neurological assessments.** This figure illustrates (a) Comparison of overall accuracy of various models. (b) Performance across three distinct dimensions of question complexity. (c) Radar charts comparing the performance accuracy across various neurological subspecialties.

## RAG framework enhancement

Building on our base model findings, we implemented a specialized RAG framework to address identified limitations in model performance to support fine-grained retrieval of relevant clinical information (**Methods**). This approach allowed models to access current clinical guidelines, detailed pathophysiological explanations, and standardized treatment protocols during their reasoning process.

The RAG enhancement demonstrated variable effectiveness across models, with benefits inversely related to base model performance (Fig 3a and Table 1). LLaMA 3.3-70B saw its accuracy increase from 69.5% (95% CI: 64.1-74.4%) to 73.4% (95% CI: 68.2-78.1%), while GPT-4o performance improved from 80.5% (95% CI: 75.9-84.7%) to 87.3% (95% CI: 83.0-90.5%). Smaller models showed the most dramatic relative improvements, with DeepSeek-R1-8B increasing

**Table 1. Performance comparison of base models, RAG enhancement, and agent-based approaches.**

| Model | Base | | RAG Enhanced | | | Agentic Framework | | |
|---|---|---|---|---|---|---|---|---|
| | Accuracy (95% CI) | F1 | Accuracy (95% CI) | F1 | P-Value | Accuracy (95% CI) | F1 | P-value |
| o1 | 90.9% (87.1-93.6) | 0.952 | 92.2% (88.6-94.7) | 0.959 | 0.664 | 94.6% (91.6-96.7) | 0.973 | 0.085 |
| GPT-4o | 80.5% (75.9-84.7) | 0.892 | 87.3% (83.0-90.5) | 0.932 | 0.027 | 89.3% (85.2-92.2) | 0.943 | 0.003 |
| LLaMA 3.3-70B | 69.5% (64.1-74.4) | 0.82 | 73.4% (68.2-78.1) | 0.846 | 0.326 | 89.2% (85.2-92.2) | 0.943 | 0 |
| DeepSeek-R1-70B | 87.7% (83.4-90.8) | 0.934 | 86.7% (82.3-89.9) | 0.929 | 0.809 | – | – | – |
| GPT-4o-mini | 60.7% (55.1-66.0) | 0.756 | 73.4% (68.2-78.1) | 0.846 | 0.001 | – | – | – |
| OpenBioLLM-70B | 65.9% (60.4-71.0) | 0.795 | 68.8% (63.4-73.8) | 0.815 | 0.491 | – | – | – |
| Meditron-70B | 52.9% (47.2-58.3) | 0.692 | 41.2% (35.9-46.9) | 0.584 | 0.004 | – | – | – |
| LLaMA 3.1-8B | 57.5% (51.8-62.8) | 0.73 | 65.9% (60.4-71.0) | 0.795 | 0.038 | – | – | – |
| DeepSeek-R1-8B | 46.8% (41.4-52.5) | 0.637 | 67.9% (62.4-72.9) | 0.809 | 0 | – | – | – |
| Medllama3 v20 | 46.8% (41.4-52.5) | 0.637 | 54.9% (49.1-60.2) | 0.709 | 0.053 | – | – | – |

This table presents the performance metrics of various language models across three evaluation scenarios. Performance is measured by accuracy with 95% confidence intervals (Wilson score method) and F1 scores. P-values (Fisher's exact test) indicating statistical significance compared to base performance. P-values are unadjusted for multiple comparisons and should be interpreted as exploratory. Subspecialty analysis revealed notable variations in model performance (Table C in S1 File). The top-performing o1 model demonstrated remarkable consistency across subspecialties, achieving 100% accuracy in several domains including headache and dizziness, neuroimmunology, cerebrospinal fluid (CSF) circulation disorders, and epilepsy. In contrast, LLaMA 3.3-70B showed significant performance variability across subspecialties. While it performed relatively well in headache and dizziness (86%) and neuro-oncology (78%), it struggled considerably with CSF circulation disorders (44%) and vascular neurology (64%). This disparity suggests that LLaMA 3.3-70B lacks the specialized knowledge or reasoning capabilities required for certain neurological domains. Interestingly, even the generally strong o1 model showed relative weakness in genetic neurology (85%) and neurophthalmology (89%), indicating that these subspecialties may present inherent challenges due to their requirements for detailed visual-spatial reasoning or complex pattern recognition.

from (95% CI: 41.4-52.5%) to 67.9% (95% CI: 62.4-72.9%) and LLaMA 3.1-8B rising from 57.5% (95% CI: 51.8-62.8%) to 65.9% (95% CI: 60.4-71.0%),. In contrast, the highest-performing models showed minimal benefit; The OpenAI-o1 model increased only from 90.9% (95% CI: 87.1-93.6%) to 92.2% (95% CI: 88.6-94.7%), reflecting ceiling effects from its already strong baseline performance. Notably, Meditron-70B showed a performance decline with RAG, 52.9% (95% CI: 47.2-58.3%) to 41.2% (95% CI: 35.9-46.9%), suggesting potential conflicts between its domain-specific training and the retrieved information.

Domain-specific analysis revealed that RAG's impact on LLaMA 3.3-70B was inconsistent across neurological subspecialties (Table C in S1 File). While significant improvements occurred in CSF circulation disorders (44% to 89%) and neurophthalmology (44% to 72%), many subspecialties showed only marginal gains or no improvement at all. Most notably, performance on headache and dizziness questions declined (100% to 50%, though only 4 questions), and areas like vascular neurology (64% to 68%) and neuroimmunology (63% to 72%) showed only modest improvements. Contrary to our hypothesis, simply providing access to specialized neurological knowledge through RAG was insufficient to substantially enhance LLaMA's performance across the full spectrum of neurological subspecialties.

When analyzed through our complexity framework, the RAG enhancement showed limited improvements for the most challenging questions (Fig 3b). For LLaMA 3.3-70B, RAG provided some improvements across complexity dimensions but fell short on level 3 complexities: FKD L3 improved from 57.5% to 70.0%, CCI L3 from 66.3% to 70.2%, and RC L3 from 61.0% to 73.2% – still leaving substantial room for improvement. The o1 model showed minimal gains across complexity levels, with improvements primarily in CCI L3 (86.5% to 89.4%) while showing no improvement in FKD L3 (87.5%). GPT-4o showed some improvements in FKD L3 (70.0% to 85.0%) and RC L3 (80.5% to 87.8%).

These complexity-based results reveal a key insight: while RAG provides some benefit for questions requiring specialized knowledge (FKD L3) and advanced reasoning (RC L3), the improvements are inconsistent and often insufficient for the most challenging neurological questions. The varying efficacy of RAG across models also indicates that even

models with stronger reasoning capabilities cannot fully overcome fundamental limitations by simply accessing external knowledge.

When evaluated on the MedQA neurological questions subset, RAG enhancement showed a more varied pattern of performance changes (Table D in S1 File). GPT-4o saw a notable increase from 85.2% (95% CI: 78.7-89.9%) to 89.7% (95% CI: 83.9-93.5%), while LLaMA 3.3-70B showed a slight decline from 76.8% (95% CI: 69.5-82.7%) to 74.8% (95% CI: 66.8-80.4%). The o1 model also demonstrated a slight decrease in performance(96.8% (95% CI: 92.7-98.6%) to 94.8% (95% CI: 89.3-96.9%)).

**Multi-agent framework results**

To address the limitations of both base models and standard RAG approaches, we developed a multi-agent framework specifically designed for complex neurological question-answering (Fig 1). Our multi-agent framework achieved remarkable performance improvements, particularly for models that had shown more modest gains with standard RAG. The LLaMA 3.3-70B-based agentic system reached 89.2% accuracy (95% CI: 85.2-92.2%, p<0.001 compared to base model), representing a dramatic 19.7 percentage point improvement over its base performance and a substantial 15.8 percentage point gain over its RAG-enhanced version. The GPT-4o implementation achieved 89.3% accuracy (95% CI: 85.2-92.2%, p=0.003 vs base), while the OpenAI-o1-based system reached 94.6% (95% CI: 91.6-96.7%) accuracy (p=0.085 vs base) (Fig 3c and Table 1).

Most striking was the effectiveness in addressing the highest complexity neurological questions that had remained challenging even after RAG enhancement. For LLaMA 3.3-70B, the multi-agent approach dramatically improved performance on Level 3 complexity questions across all dimensions: FKD L3 increased from 70.0% (RAG) to 87.5% (agent), CCI L3 from 70.2% to 90.3%, and RC L3 from 73.2% to 92.6% (Fig 3b). This pattern of improvement on the most complex questions was consistent across models, though less dramatic for o1 given its already strong performance.

The multi-agent approach also addressed LLaMA 3.3-70B's inconsistent subspecialty performance that persisted even with RAG enhancement. With the agentic framework, performance became remarkably consistent across neurological subspecialties, with substantial improvements in previously challenging areas like headache and dizziness (50% with RAG to 100% with agent), neuromuscular disorders (74% to 100%), and neuroimmunology (67% to 86%). The improvement in neurophthalmology (72% to 93%) and movement disorders (81% to 94%) was also substantial. This consistent cross-specialty performance suggests that the multi-agent architecture effectively overcomes the domain knowledge limitations inherent in the base model (Fig 3c).

---

**Box 1. Step-by-step analysis of multi-agent framework performance on a complex clinical question.**

**Question:** A 70-year-old man undergoing peritoneal dialysis for end-stage diabetic kidney failure develops progressive limb weakness over two weeks, accompanied by paresthesias in all four limbs. There is no history of fever or vaccination preceding the illness. Nerve conduction studies show significant prolongation of motor and sensory latencies, unevenly distributed, without conduction blocks or dispersion. CSF analysis reveals no cells, protein 55 mg/dL, and normal glucose. There is no significant improvement after two full courses of IVIG. Which of the following could lead to the patient's recovery?

**Choices:** a. Kidney transplantation | b. Another course of IVIG | c. Hemodialysis | d. Steroids

**Correct Answer:** a | **Framework Answer:** a

**Agent 1 - Question Complexity Classifier:** Classified as "complex," requiring integration of clinical findings, treatment history, comorbid conditions, and therapeutic options rather than simple factual recall.

---

**Agent 2 - Question Interpreter:** Extracted key concepts: progressive weakness, IVIG failure, end-stage renal disease on dialysis, albuminocytologic dissociation. Generated five retrieval queries targeting treatment effectiveness:

- "uremic neuropathy in end-stage diabetic kidney failure progressive weakness no response to IVIG role of kidney transplant for recovery"
- "progressive demyelinating neuropathy diabetic ESRD peritoneal dialysis improving nerve conduction after kidney transplant"
- "CIDP versus uremic neuropathy in diabetic renal failure and potential role of kidney transplant"

**Agent 3 - Research Retrieval:** Retrieved key evidence: "Successful renal transplantation results in significant clinical, EDX, and morphological recovery over a period of 3-12 months" but "renal transplantation may have little effect on the course of the polyneuropathy in diabetic patients with end-stage renal disease." Additional passages: "Chronic peritoneal dialysis provides no advantage over hemodialysis" and IVIG-nonresponsive cases rarely benefit from repeated courses.

**Agent 4 - Answer Synthesis:** Evaluated each option systematically. Selected kidney transplantation (a) reasoning: "the primary factor aggravating the patient's neuropathy is likely his inadequate clearance of toxins and metabolic derangements from end-stage renal disease; kidney transplantation can often reverse or markedly improve such neuropathies." Rejected other options: "Repeating IVIG, changing dialysis modality, or adding steroids is less likely to be effective unless the underlying renal failure is corrected."

**Agent 5 - Validator:** Confirmed clinical features matched uremic neuropathy rather than inflammatory conditions (no conduction blocks, IVIG failure, elevated CSF protein without pleocytosis). Validated that kidney transplantation "addresses the underlying metabolic derangements causing neuropathy" while "repeated immunotherapy or dialysis changes alone are unlikely to result in full recovery." Approved answer.

This box illustrates how the five specialized agents collaborate to analyze a complex clinical question, tracing the systematic process from initial complexity classification and concept extraction, through knowledge retrieval and answer synthesis, to final validation of the reasoning. To illustrate how the multi-agent architecture handles therapeutic reasoning, Box 1 presents a representative example involving treatment selection for uremic neuropathy in a dialysis patient unresponsive to IVIG. The Question Interpreter generated queries targeting treatment effectiveness rather than diagnostic patterns (e.g., "role of kidney transplant for recovery in IVIG-nonresponsive uremic neuropathy"). The Research Retrieval agent gathered evidence on transplantation outcomes, including the critical distinction that "renal transplantation results in significant clinical recovery" in uremic neuropathy but "may have little effect in diabetic patients with end-stage renal disease." The Answer Synthesis agent systematically evaluated each option, selecting kidney transplantation based on its potential to correct the underlying metabolic derangements rather than providing symptomatic management. The Validator confirmed this reasoning by verifying that clinical features matched uremic rather than inflammatory neuropathy, approving kidney transplantation as addressing the root cause.

The multi-agent framework effectiveness was further validated on the MedQA neurological question subset (Table E in S1 File). The LLaMA 3.3-70B-based agentic system achieved 81.3% accuracy (95% CI: 74.4-86.6%) on MedQA questions, representing a 4.5 percentage point improvement over its base performance (76.8%, 95% CI: 69.5-82.7%). GPT-4o's agentic implementation reached 94.8% (95% CI: 89.3-96.9%) on the MedQA questions, a 9.6 percentage point increase from its base performance (85.2%, 95% CI: 78.7-89.9%), with o1 maintaining excellent performance at 94.8% (95% CI: 89.3-96.9%).

### Error analysis of multi-agent framework

To understand the limitations of our multi-agent framework, we conducted a systematic analysis of the remaining errors made by the top-performing model. The framework's errors fall into two primary categories: retrieval failures (RAG gaps where relevant information was not retrieved from the knowledge base) and reasoning failures (where retrieved information was present but misused or ignored). Within reasoning failures, we identified four distinct patterns: (1) ignored explicit evidence - correct answers existed in retrieved text but were disregarded in favor of alternative reasoning; (2) narrative distraction - construction of sophisticated but incorrect explanations instead of matching straightforward clinical patterns; (3) over-integration - forcing disparate clinical findings into a single diagnosis when they belonged to different conditions; and (4) pattern mismatching - failure to connect clinical descriptions in questions to corresponding descriptions in retrieved passages.

Critically, validation loops failed differently across error types. For retrieval gaps, validation agents had no mechanism to detect missing information, as they could only assess the coherence of reasoning based on available evidence. For reasoning failures, validation loops often reinforced initial errors rather than catching them, suggesting the quality assurance agent shared the same reasoning blind spots as the synthesis agent—both prioritizing internal narrative coherence over strict textual fidelity to retrieved evidence. Representative examples demonstrating each error pattern with detailed analysis are provided in S3 File.

### Discussion

This study makes three key contributions to the field of AI in clinical neurology: creating a comprehensive neurological assessment benchmark derived from board certification examinations, systematically evaluating current LLM capabilities, and developing a novel multi-agent framework specifically designed for complex neurological reasoning.

Our benchmark provides a valuable addition to the landscape of AI evaluation frameworks by focusing specifically on the advanced reasoning patterns required in neurological practice. Board certification questions across 13 neurological subspecialties capture the intricate reasoning patterns that characterize expert practice, offering a pragmatic measure of whether AI systems can approach board-certified specialists' reasoning levels. Our three-dimensional complexity framework - factual knowledge depth, clinical concept integration, and reasoning complexity - revealed a moderate correlation between these dimensions (0.51-0.67), confirming they measure distinct aspects of neurological reasoning. The complexity classification framework reveals the distinctive cognitive demands of neurological assessment. Questions requiring the integration of multiple clinical concepts and sophisticated reasoning were prevalent throughout the benchmark, reflecting the reality that neurological diagnosis frequently relies on synthesizing seemingly disparate symptoms and signs into coherent clinical syndromes.

Our evaluation reveals a clear performance hierarchy, with the reasoning-specialized models (OpenAI-o1: 90.9%) outperforming general-purpose models (GPT-4o: 80.5%), while medical domain-specialized models underperformed (OpenBioLLM-70B: 65.9%, Meditron-70B: 52.9%). All models showed performance degradation as complexity increases, particularly for questions requiring level 3 complexity in clinical concept integration and factual knowledge depth. Even top-performing models showed relative weaknesses in certain domains, highlighting the uneven development of current AI systems and validating the need for specialized approaches to enhance model performance in clinical contexts. Our findings contribute context to recent discussions about the suitability of LLMs for clinical decision-making [28]. While base model performance shows promising capabilities, the significant degradation on higher complexity questions confirms that unmodified LLMs still face important limitations when handling the most challenging neurological cases. This validates the need for specialized approaches to enhance model performance in clinical contexts, particularly for subspecialties requiring the integration of multiple clinical concepts and sophisticated temporal reasoning.

Our RAG implementation demonstrated improvements inversely proportional to base model strength—dramatic for smaller models (DeepSeek-R1-8B: 46.8% to 67.9%), substantial for mid-tier models (GPT-4o: 80.5% to 87.3%), modest for larger models (LLaMA 3.3-70B: 69.5% to 73.4%), and minimal for the highest-performing model (o1: 90.9% to 92.2%).

This pattern suggests that while RAG can effectively address knowledge gaps, simply providing specialized knowledge access is insufficient for overcoming the fundamental reasoning challenges in complex neurological cases. Particularly striking was RAG's limited effectiveness on level 3 complexity questions across all dimensions, suggesting that the most challenging aspects of neurological reasoning require more sophisticated frameworks. Furthermore, the divergent RAG effects between board certification questions (consistent improvements) and MedQA questions (minimal or negative impact) highlight a critical consideration: RAG performance depends heavily on alignment between the knowledge base and target assessment domain. Our specialized neurology textbook enhanced reasoning on neurological board examinations but provided limited value for MedQA's broader medical questions, indicating that clinical AI systems may require multiple, complementary knowledge bases to handle the full spectrum of medical reasoning tasks encountered in practice.

Our multi-agent framework addresses these limitations by decomposing complex neurological reasoning into specialized cognitive functions distributed across five distinct agents (Fig 1). This approach yielded dramatic improvements, particularly for LLaMA 3.3-70B (69.5% to 89.2%), demonstrating structured reasoning's power for complex clinical tasks. Most notably, the framework excelled at the highest complexity questions that had remained challenging even after RAG enhancement, with LLaMA 3.3-70B showing substantial improvements across all dimensions (FKD L3: 70.0% to 87.5%, CCI L3: 70.2% to 90.3%, RC L3: 73.2% to 92.6%). The framework also transformed previously inconsistent subspecialty performance into remarkably uniform excellence, effectively addressing domains that had proven difficult for both base models and RAG-enhanced implementations.

Across all implementations, the OpenAI-o1 model demonstrated superior capabilities, with our o1-based agentic system achieving near-perfect performance. Upon detailed analysis of the few remaining errors, we found that many involved questions where the correct answer was not entirely clear-cut, with potential ambiguities in the question formulation or cases where multiple approaches could be justified depending on specific clinical circumstances.

External validation using 155 neurological cases from the MedQA yielded two key findings: Models performed better on MedQA than on board certification questions, confirming our benchmark's higher complexity; and enhancement strategies showed dataset-specific patterns. RAG improved performance on board questions but showed limited impact on MedQA, with smaller gains for GPT-and a slight decrease for LLaMA 3.3-70B. This discrepancy likely stems from our RAG system using a specialized neurology textbook that aligned with board questions but less with MedQA questions requiring broader medical knowledge. Commercial models like GPT-4o were less affected due to their broader knowledge base. These findings emphasize the importance of aligning knowledge resources with specific information needs when implementing RAG and agent-based systems.

Our findings extend recent work by Schubert et al. [7], who demonstrated GPT-4's capability to exceed mean human performance on neurology board examinations (85.0% vs 73.8%). While they identified performance degradation with increasing question complexity, our multi-agent framework maintained remarkably consistent performance across all complexity levels. Similarly, compared to Masanneck et al. [29], who showed modest improvements through standard RAG approaches, our multi-agent framework achieves substantially greater performance gains, particularly for complex questions requiring advanced clinical concept integration. This comparison highlights the advantage of specialized cognitive frameworks over simple knowledge retrieval for complex neurological reasoning tasks. Furthermore, our three-dimensional complexity classification provides deeper insights into specific reasoning capabilities than the binary knowledge-based versus case-based categorization used in their work. These complementary findings collectively suggest that advancing AI applications in neurology will require both robust knowledge integration and sophisticated reasoning architectures that mirror the structured problem-solving approaches used by clinical experts.

Several limitations must be considered: our benchmark excludes visual elements crucial in neurological assessment; strong performance on board-style questions may not fully translate to real-world clinical scenarios with incomplete information; our RAG system showed dataset-specific effectiveness; and our multi-agent framework's computational requirements may challenge real-time clinical application.

Looking forward, this work suggests promising research directions: further development of specialized architectural components for clinical reasoning, more sophisticated knowledge integration approaches, and extending these

approaches to handle multimodal inputs crucial to neurological assessment. While current models show promising capabilities, their limitations in complex reasoning and management decisions suggest they are better suited for supporting rather than replacing clinical decision-making. The success of structured approaches like our multi-agent framework suggests that future clinical AI systems might be most effective when designed to complement and enhance human clinical reasoning rather than replicate it entirely.

## Supporting information

**S1 File. Table A.** Comparison of LLMs and RLMs specifications and deployment characteristics. Table B. Distribution of neurology subspecialties across different exams. Table C. Accuracy analysis of base, RAG enhanced, and agentic methods across neurological subspecialties. Table D. Comparative performance on board certification (N = 305) and MedQA neurological questions (N = 155). Table E. Accuracy analysis of base, RAG enhanced, and agentic methods across neurological subspecialties in exam 1062023. Table F. Accuracy analysis of base, RAG enhanced, and agentic methods across neurological subspecialties in exam 1052024. Table G. Accuracy analysis of base, RAG Enhanced, and agentic methods across neurological subspecialties in exam 1092024.
(DOCX)

**S2 File. Three-dimensional complexity classification of 305 neurological board certification questions by factual knowledge depth, clinical concept integration, reasoning complexity and reasoning type.**
(DOCX)

**S3 File. Systematic error analysis of multi-agent framework performance on six representative questions demonstrating retrieval failures and reasoning failures across different neurological subspecialties and complexity levels.**
(DOCX)

## Author contributions

**Conceptualization:** Moran Sorka, Dvir Aran, Shahar Shelly.

**Data curation:** Alon Gorenshtein, Shahar Shelly.

**Formal analysis:** Moran Sorka.

**Investigation:** Moran Sorka.

**Methodology:** Moran Sorka, Dvir Aran, Shahar Shelly.

**Software:** Moran Sorka.

**Supervision:** Dvir Aran.

**Visualization:** Moran Sorka.

**Writing – original draft:** Moran Sorka, Dvir Aran, Shahar Shelly.

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
