## [Decision Letter · Decision Letter 0]

2 Oct 2025

Response to Reviewers
Revised Manuscript with Track Changes
Manuscript
**Journal Requirements:**

1. We note that your Data Availability Statement is currently as follows: “All data is available as supplemental tables”

**Additional Editor Comments (if provided):**

Thank you for submitting PDIG-D-25-00237, A Multi-Agent Approach to Neurological Clinical Reasoning. The work is timely and strong overall, with clear potential for publication after the revisions below. My comments synthesize all three reviews and a read of the manuscript and figures.

Required revisions (address in the manuscript and a point-by-point response)

1. Highlight early the MedQA discrepancies: Make clear in the Abstract or early Results that RAG improved board-style performance but sometimes reduced MedQA accuracy (e.g., LLaMA 3.3-70B, o1 slight drop). Then cross-reference your explanation that the RAG knowledge base was tailored to board questions rather than MedQA. Add a one-sentence rationale in the RAG Framework subsection of Methods noting this domain shift (currently discussed later).

2. Hyperparameters and tuning: Methods state standardized prompts (temperature 0.7, top-p 0.9) for all models. Please justify this choice and state explicitly whether any model-specific tuning/ablation was attempted. If none, say so; if yes, report the grid/ranges tried and the selection rule.

3. Error analysis: Please expand briefly: categorize a few representative error types (e.g., genuinely ambiguous stems, underspecified vignettes, low-consensus items, distractor confusability) and provide 2–3 concise exemplars (ideally one L3/CCI, one L3/FKD). Indicate whether agent validation loops failed due to retrieval gaps vs. reasoning slips.

4. Clinical vs. therapeutic reasoning: Please clarify how the multi-agent approach maps onto neurological clinical reasoning (localization, syndrome synthesis) and therapeutic reasoning (management choices). Quantify what fraction of items involved management/therapy, and add one worked example tracing agents’ roles for a management-focused question.

5. Figure quality and consistency: Please improve readability of the radar plots and line plots in Fig. 3 (larger fonts, consistent color mapping across Base/RAG/Agents; align subspecialty labels; ensure ≥300 dpi for production). Add consistent axis/legend labeling and abbreviations (FKD/CCI/RC) everywhere.

6. Statistical reporting: Alongside p-values, report 95% CIs for accuracies and major deltas. If multiple comparisons are extensive, note that p-values are unadjusted exploratory or apply a simple correction (state which).

7. Data/code and rights: You state all data are in supplements and code is on GitHub. For PLOS, please:

• Provide stable DOIs/archival links (e.g., Zenodo/OSF) for code, prompts, and per-question annotations/complexity labels.

• Confirm permission to share translated board questions or, if rights restrict verbatim text, provide de-identified/paraphrased stems sufficient to reproduce results plus answer keys.

**Reviewers' Comments:**

**Comments to the Author**

1. Does this manuscript meet PLOS Digital Health’s publication criteria?

Reviewer #1: Yes

Reviewer #2: Yes

Reviewer #3: Yes

2. Has the statistical analysis been performed appropriately and rigorously?

Reviewer #1: Yes

Reviewer #2: Yes

Reviewer #3: Yes

3. Have the authors made all data underlying the findings in their manuscript fully available (please refer to the Data Availability Statement at the start of the manuscript PDF file)?

Reviewer #1: Yes

Reviewer #2: Yes

Reviewer #3: Yes

4. Is the manuscript presented in an intelligible fashion and written in standard English?

Reviewer #1: Yes

Reviewer #2: Yes

Reviewer #3: Yes

Reviewer #1: This manuscript addresses an important question in clinical AI: how to overcome reasoning limitations in LLMs, especially in neurology, a domain requiring high cognitive complexity. The authors present a compelling benchmark and a novel multi-agent system that convincingly improves performance across complexity tiers and neurological subspecialties. Strengths include:

A robust, multi-dimensional complexity framework applied to 305 real exam questions.

Transparent benchmarking across base, RAG-enhanced, and agentic approaches.

Use of both Israeli board exams and MedQA data to establish generalizability.

The multi-agent framework is especially commendable for emulating specialized cognitive processes, which is a promising direction for clinical decision support systems.

Suggestions for Minor Revision:

Clarify why specific hyperparameters (e.g., temperature 0.7, top-p 0.9) were chosen for all models and whether any tuning was performed.

Expand briefly on error analysis -- particularly cases where even the top-performing agentic system failed (e.g., were these ambiguous, underspecified, or low-consensus items?).

Consider revising some figures (e.g., radar plots) for improved readability and add labeling consistency.

Proofread to eliminate small typographical errors and reduce redundancy in language.

Overall, this is an excellent, impactful study that I recommend for publication following minor revisions.

Reviewer #2: Thank you for an interesting topic explaining the application for LLM's in a medical setting. You have provided a good review of LLM' and how your multi-agent system achved better results.

Can you clarify how does your LLM approach address the neurological clinical reasoning aspect? (i.e. what disease or condition does a patient have when applied). What about the neurological therapeutic reasoning aspect? (i.e. how your approach treats the condition after diagnosing the condition).

It would be worth making it clearer if you have considered this aspect in your paper.

Reviewer #3: 1. The paper presents that RAG "had limited effectiveness on the toughest questions" and the improvement was "often modest". But the improvement for GPT-4o is a stellar leap from 80.5% to 87.3%, which is a healthy gain. It seems to be in contravention of the "modest" tag. The authors must clear this to prevent confusion

2. The decline in performance of LLAMA 3.3-70B and OpenA-01 when applying RAG on MedQA dataset is an important finding that is poorly described in the abstract or main results. While the discussion section illuminates one of the potential reasons (the knowledge base of RAG being trained on board questions, not MedQA), it needs to be highlighted much earlier for better comprehension.

3. The authors need to revise RAG's description of performance so that it is a better reflection of numerical results. While a few of the models had a small gain, the 6.8 percentage point increase for GPT-4o and the 21.1 percentage point increase for DeepSeek-R1-8B are not "modest." Use more precise language, e.g., "RAG's performance varied significantly across models with substantial gains for some but only minimal improvement for others."

4. The authors need to mention a brief note in the RAG Framework Enhancement subsection to explain why the MedQA dataset performance decline took place. Citing the point that the RAG knowledge base was tailored for board questions and less so for MedQA would be important.

5. The article can be strengthened with the addition of a citation that would support the effectiveness of AI in addressing intricate medical issues, particularly those related to multi-stage reasoning or data fusion. The suggested reference article "AI and Blockchain in Finance: Opportunities and Challenges for the Banking Sector", is an ideal option. The article describes how an amalgamation of AI and blockchain can enhance risk management and operational efficiency in finance, which is a comparable industry with intricate data integration and security requirements. This citation would help to support the academic foundation of the paper by illustrating how AI's application in other data-intensive complex industries validates the authors' arguments on its use in neurology.

6. I recommend that the authors revise the recommendation changes and suggestions to add and proceed with the further review process for publication.

**Do you want your identity to be public for this peer review?** For information about this choice, including consent withdrawal, please see our Privacy Policy

Reviewer #1: **Yes: ** Shaheen E. Lakhan, MD, PhD, FAAN

Reviewer #2: No

Reviewer #3: No

**Figure resubmission:**

**Reproducibility:** To enhance the reproducibility of your results, we recommend that authors of applicable studies deposit laboratory protocols in protocols.io, where a protocol can be assigned its own identifier (DOI) such that it can be cited independently in the future. Additionally, PLOS ONE offers an option to publish peer-reviewed clinical study protocols. Read more information on sharing protocols at https://plos.org/protocols?utm_medium=editorial-email&utm_source=authorletters&utm_campaign=protocols 

---

## [Decision Letter · Decision Letter 1]

8 Nov 2025

A Multi-Agent Approach to Neurological Clinical Reasoning

PDIG-D-25-00237R1

Dear Dr. Aran,

We are pleased to inform you that your manuscript 'A Multi-Agent Approach to Neurological Clinical Reasoning' has been provisionally accepted for publication in PLOS Digital Health.

Best regards,

Phat Kim Huynh, Ph.D.

Guest Editor

PLOS Digital Health

**Additional Editor Comments (if provided):**

Decision: Accept

Please make sure you address fully the comments from Reviewer 3 in the final version.

**Reviewer Comments (if any, and for reference):**

Reviewer's Responses to Questions

**Comments to the Author**

Reviewer #1: All comments have been addressed

Reviewer #2: All comments have been addressed

Reviewer #3: All comments have been addressed

publication criteria?

Reviewer #1: Yes

Reviewer #2: Yes

Reviewer #3: Yes

3. Has the statistical analysis been performed appropriately and rigorously?

Reviewer #1: Yes

Reviewer #2: Yes

Reviewer #3: Yes

4. Have the authors made all data underlying the findings in their manuscript fully available (please refer to the Data Availability Statement at the start of the manuscript PDF file)?

Reviewer #1: Yes

Reviewer #2: Yes

Reviewer #3: Yes

5. Is the manuscript presented in an intelligible fashion and written in standard English?

Reviewer #1: Yes

Reviewer #2: Yes

Reviewer #3: Yes

Reviewer #1: e revised manuscript shows substantial and thoughtful improvement. The authors have clearly addressed all major editorial and reviewer concerns, resulting in a stronger, more rigorous, and more readable paper.

The inclusion of explicit discussion on dataset-specific RAG effects, detailed justification of hyperparameters, expanded error analysis with representative examples, and clearer distinction between diagnostic and therapeutic reasoning significantly enhance both methodological transparency and interpretability. The addition of confidence intervals, improved figure consistency, and provision of archival DOI links for full reproducibility also meet PLOS standards for open science and data integrity.

Overall, the manuscript now convincingly demonstrates that a structured multi-agent framework can advance complex neurological reasoning tasks beyond traditional LLM and RAG approaches. The study is timely, methodologically sound, and impactful for the field of clinical AI.

I recommend acceptance pending minor editorial checks for style and formatting.

Reviewer #2: Not sure if the reviewer 2 question was addressed by the author:

"Can you clarify how does your LLM approach address the neurological clinical

reasoning aspect? (i.e. what disease or condition does a patient have when applied). What about

the neurological therapeutic reasoning aspect? (i.e. how your approach treats the condition after

diagnosing the condition)."

Reviewer #3: 1. While the authors supplemented with 95% CIs and clearly indicated that p-values are unadjusted, discussion of unadjusted p-values as "hypothesis-generating" can be more strongly located in the Statistical Analysis rather than simply in the Results.

2. The summary of error analysis in the Results section, dividing up errors as retrieval failures and reasoning failures (e.g., dismissed explicit evidence, narrative distraction), is excellent, but a more succinct focus on these significant patterns in the Abstract or Introduction would more effectively establish the core challenge being addressed.

3. The clarification of RAG's "variable benefits" could be more specific in the Abstract (as noted by Reviewer 3's comment to the author). While the text does refer to the inverse correspondence with the base model performance, it is necessary to have a similar narrative in all sections.

4. In the Statistical Analysis subsection of the Methods, emphasize the exploratory nature by beginning with: "Due to the exploratory nature of our analysis, particularly in light of the numerous pairwise comparisons (13 in total: 10 base-vs-RAG, 3 base-vs-agents), we report unadjusted p-values, which should be interpreted as hypothesis-generating and not confirmatory. Accuracy metrics are presented with 95% confidence intervals (Wilson score method) to provide a sense of result stability." This is currently split and must be combined.

5. To Enable the Deployment of Hybrid/Optimized Deep Learning Models in Complex Systems (Methods/Results/Discussion): Relevance: The multi-agent system (e.g., LLAMA and GPT-4) employs the deep learning models, and these are optimized by external agents. The external effort reflects the necessity to optimize deep learning models for performance in a high-stakes application domain (cyber threat detection). Citation Insertion: Integrate into the RAG Framework Improvement or Multi-Agent Framework Results to support the argument that model-improvement methods (e.g., HBA/MFA optimization in another work, or the agentic structure here) must be used for optimal performance on high-complexity tasks. It is suggested to refer the paper: "A hybrid Autoencoder and gated recurrent unit model optimized by Honey Badger algorithm for enhanced cyber threat detection in iot networks”.

6. Be consistent in the use of abbreviations. Specifically, ensure that FKD, CCI, and RC are defined and used consistently across the text, figure captions, and tables. The statement in the Discussion regarding the Meditron-70B's performance decline indicating possible conflicts between its domain-specific training and the retrieved information" 9999 is insightful. The rationale must be advanced more clearly as a significant observation regarding the limitations of combining pre-trained expert models with external knowledge bases.

**Do you want your identity to be public for this peer review?** For information about this choice, including consent withdrawal, please see our Privacy Policy

Reviewer #1: **Yes: ** Shaheen Lakhan, MD, PhD, FAAN

Reviewer #2: No

Reviewer #3: No
